# Tailoring tokamak error fields to control plasma instabilities and transport

SeongMoo Yang [1] ✉, Jong-Kyu Park [1,2], YoungMu Jeon [3], Nikolas C. Logan[4], Jaehyun Lee[3], Qiming Hu[1], JongHa Lee[3], SangKyeun Kim[1], Jaewook Kim [3], Hyungho Lee[3], Yong-Su Na [2], Taik Soo Hahm[2], Gyungjin Choi [2], Joseph A. Snipes [1], Gunyoung Park[3] & Won-Ha Ko[3]

A tokamak relies on the axisymmetric magnetic fields to confine fusion plasmas and aims to deliver sustainable and clean energy. However, misalignments arise inevitably in the tokamak construction, leading to small asymmetries in the magnetic field known as error fields (EFs). The EFs have been a major concern in the tokamak approaches because small EFs, even less than 0.1%, can drive a plasma disruption. Meanwhile, the EFs in the tokamak can be favorably used for controlling plasma instabilities, such as edge-localized modes (ELMs). Here we show an optimization that tailors the EFs to maintain an edge 3D response for ELM control with a minimized core 3D response to avoid plasma disruption and unnecessary confinement degradation. We design and demonstrate such an edge-localized 3D response in the KSTAR facility, benefiting from its unique flexibility to change many degrees of freedom in the 3D coil space for the various fusion plasma regimes. This favorable control of the tokamak EF represents a notable advance for designing intrinsically 3D tokamaks to optimize stability and confinement for next-step fusion reactors.

Unlike the sun, which uses gravity to confine fusion plasmas, a tokamak uses a strong magnetic field for plasma confinement aiming to generate clean energy without carbon emission and long-lived radioactive waste. Since fusion reactions occur most effectively at high temperatures of more than 100 million degrees, the biggest challenge for the tokamak is to improve its stability and confinement to maintain high-performance fusion plasmas.

Tokamaks can sustain higher-performance fusion plasmas with their toroidally axisymmetric magnetic field, unlike other magnetic fusion concepts[1–3]. However, small non-axisymmetric (3D) error fields (EFs) inevitably arise in tokamaks from the imperfections and misalignments of magnetic field coils[4]. Eliminating the intrinsic EFs from the complex tokamak system is highly challenging, requiring extensive resources and assembly time. As small-level EFs of less than 0.1% can lead to plasma disruption[5,6], and confinement degradation[4,7], reducing the 3D EFs has been a longstanding and vital concern in tokamak construction, especially for the multibillion-dollar project ITER ('The

Way' in Latin). Simultaneously with extensive efforts to reduce the 3D EFs, ITER has adopted a small externally applied 3D field to prevent excessive material erosion[8,9] due to edge instabilities known as edge localized modes (ELMs)[10,11]. The externally applied 3D field is called resonant magnetic perturbation (RMP), and it has been a widely accepted approach to prevent ELMs in tokamaks to regulate the excessive increase of the edge pressure gradient[10–19]. Despite its fundamental similarity to RMP, the idea of leveraging EFs for ELM control has been almost prohibited due to their strong perturbations to the core. For the same reason, most tokamaks also employ extra 3D coils to avoid unfavorable core EF effects[4,7,20–25]. This correction of EF spectra is known as EF correction (EFC). What is missing in the standard EFC is the ability to isolate the edge from the core 3D effects, which is challenging due to the strong poloidal coupling and kink mode amplification[26–28].

The isolation of the edge 3D effects is, in fact, a goal in RMP applications as well. The current approach is to generate RMPs with high toroidal mode numbers ($n > 1$) with the in-vessel coils, relying on

[1]Princeton Plasma Physics Laboratory, Princeton, NJ 08543, USA. [2]Department of Nuclear Engineering, Seoul National University, Seoul 08826, South Korea. [3]Korea Institute of Fusion Energy, Daejeon, Republic of Korea. [4]Columbia University, New York, NY 10027, USA. ✉e-mail: syang@pppl.gov

the natural spatial decay of shorter wavelength RMPs. However, future tokamak reactors such as DEMO will need ex-vessel 3D coils to avoid nuclear degradation of the coils, leaving only a long-wavelength low-$n$ RMP as a viable option for efficient ELM control[29]. Then again, it becomes critical to systematically isolate the edge from the core 3D responses in the RMP ELM control, where the synergies with the aforementioned ability in the EFC become imperative. For example, stable ELM control using $n = 1$[11,15,30–33] and $n = 2$[34] RMPs typically requires operation within narrow windows, as an increase in the low-$n$ RMP strength also results in the emergence of locked modes in the core.

This work reports a 3D tokamak configuration correcting the longest wavelength ($n = 1$) EF while leaving edge-localized RMP (ERMP) to control both core and edge instabilities and transports in KSTAR. This ERMP is systematically given by characterizing core and edge resonant responses as coupled weakly damped oscillators. The ERMP demonstrates safe access to ELM-controlled high confinement regime (H-mode) by considering the change of the 3D plasma response from low confinement regime (L-mode) to H-mode and proves its potential to control ELMs for the entire discharge periods for various scenarios. The optimization also finds a unique direction to utilize the 3D coils with unlimited possible choices for a 3D magnetic field to access the ELM-suppressed H-mode safely and demonstrates its improved safety and confinement. Furthermore, ERMP maintains an advanced plasma regime with improved core confinement at a reactor-relevant ion temperature of 100 million kelvin by regulating edge transport of tokamak plasmas. These results show that tailoring the $n = 1$ EF of the tokamak is a promising direction to control the stability and transport of tokamaks, with the potential to improve any high-performance tokamak scenario (H-mode[35,36], I-mode[37], FIRE mode[38], negative triangularity[39,40] L-mode) to accelerate fusion science.

## Results

### Optimization of the EFC for simultaneous stabilization of core and edge instabilities

KSTAR has an advantage in investigating the design of the EFC over other tokamaks due to its low level of intrinsic $n = 1$ EF[41] and flexible three rows of the external 3D coils (Fig. 1a, b). Instead of using the intrinsic $n = 1$ EF with potential uncertainties, KSTAR can apply a known EF source using one coil row from the multiple coil rows. Throughout this work, we use the top row coil current ($I_T$) to generate a proxy $n = 1$ EF. Figure 1 shows an application of proxy EF to a typical KSTAR low-density L-mode discharge with plasma current ($I_p$) of 0.6 $MA$, toroidal field ($B_T$) of 2.0 $T$, and edge safety factor $q_{95} \sim 4.3$. As shown in Fig. 1c, the slow increase of the $n = 1$ proxy EF with the $I_T$ results in a core locked mode (LM) at $t = 3.5s$ at $I_T = 4.5kA$, identifiable by the drop of $T_e$[41]. This LM is followed by a disruption, which is indicated by the sudden decrease in the $T_e$ and the plasma current. The remaining two coil rows in KSTAR can generate an $n = 1$ field with 4-dimensional freedom, $\mathbf{I} = (I_M, I_B, \phi_{MT} = \phi_M - \phi_T, \phi_{BM} = \phi_B - \phi_M)$, where $I$ is the amplitude and $\phi$ is the phase of the $n = 1$ current distribution in either the middle (M) or bottom (B) row of coils. These two coil rows are not used in Fig. 1, but they give good flexibility for spectrum control to design and test the EFC. For a conventional $n = 1$ EFC, these two coil rows are used to minimize the core resonant coupling to prevent the disruptive core LM in the tokamak. The term "resonant" in this work represents a 3D field that resonates with an equilibrium magnetic field. More precisely, the resonant components are defined as the $m = nq$ harmonics at rational surfaces, where magnetic field lines close in on themselves after $m$ toroidal transits and $n$ poloidal transits. Here, $q$ is the so-called safety factor.

The impact of a core-to-edge optimized $n = 1$ EFC is studied in KSTAR plasmas with plasma current ($I_p$) of 0.51 $MA$, toroidal field ($B_T$) of 1.8 $T$, and $q_{95} \sim 4.3$ in L-mode and $q_{95} \sim 5.1$ in the H-mode. The $n = 1$ proxy EF is applied using the top row at the maximum current of 5 $kA$, which is typically strong enough to drive a core LM in these low-density plasmas (Fig. 1c). The other four 3D coil variables are then applied simultaneously to correct this proxy EF. The core-to-edge optimized $n = 1$ EFC is designed to prevent a core LM as in the conventional EFC, but it is also designed to suppress ELMs in H-mode by maintaining an edge resonant field.

Figure 2a shows the optimized $n = 1$ EFC applied to KSTAR experiments, clearly demonstrating its safety in the L-mode, L-H

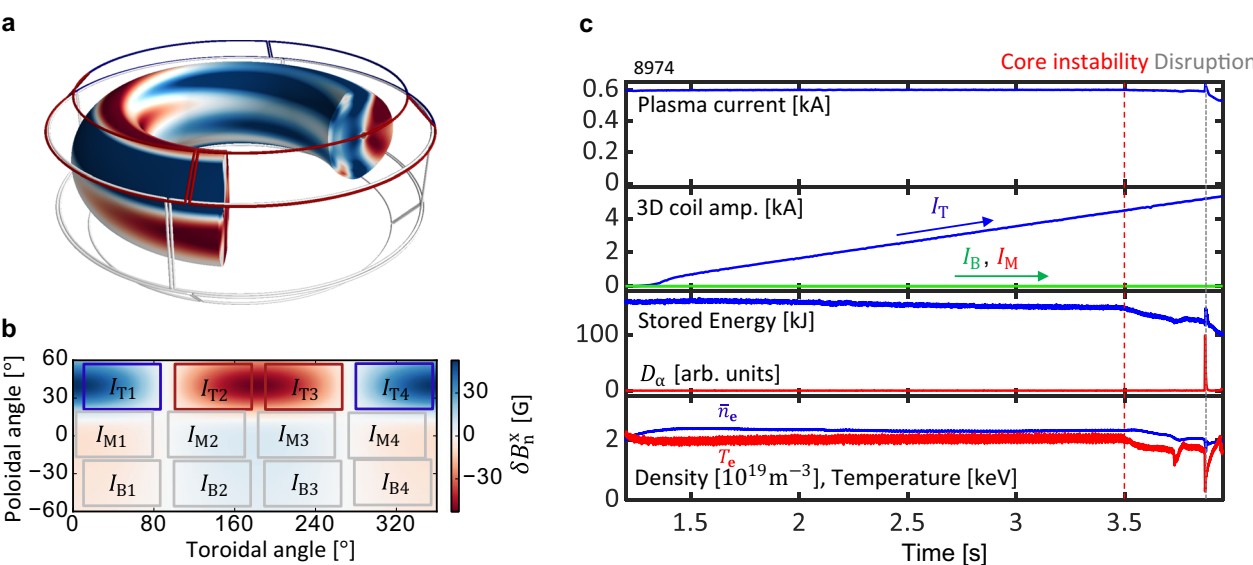

**Fig. 1 | Proxy error field applied by KSTAR 3D coils. a** The actual geometry of 3D coils and perturbed flux surface due to the plasma response (perturbation is not big enough to see the distortion of flux surface in this figure). **b** The externally applied normal magnetic field at the plasma boundary ($\delta B_n^x$) and a schematic view of 3D coils projected on the plasma boundary. **c** From top to bottom: the time evolution of the plasma current, 3D field amp., stored energy, $D_\alpha$ emission (particle recycling light) spike that shows disruptive plasma instability, line averaged density ($\bar{n}_e$), electron temperature ($T_e$) due to the slow increase of the proxy $n = 1$ error field ($I_T$) without the error field correction.

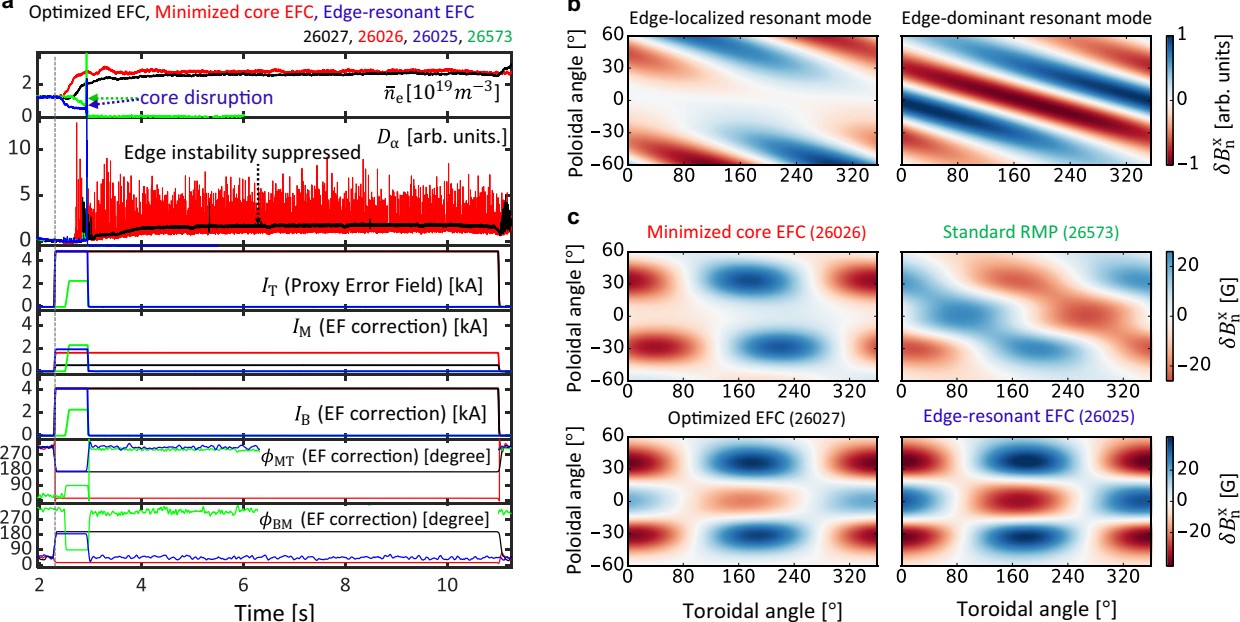

**Fig. 2 | Control of core and edge instabilities with error field correction (EFC).**
**a.** From top to bottom: the time evolution of line averaged density ($\bar{n}_e$) showing a confinement mode transition from the discharge's initial Low confinement mode (L-mode) to High confinement mode (H-mode), $D_\alpha$ emission (particle recycling light) showing a disruption, ELM crashes and ELM suppression, and 3D coil variables ($I = (I_T, I_M, I_B, \phi_{MT}, \phi_{BM})$) with optimized EFC (black), minimized core EF (red), edge resonant EFC (blue), and standard RMP (green). **b** Distribution of edge-localized and edge-dominant resonant mode on the plasma boundary ($\delta B_n^x$). **c** Externally applied normal magnetic field on the plasma boundary ($\delta B_n^x$) with the KSTAR 3D coils.

transition, and H-mode phases as well as its ability to suppress ELMs in the H-mode phase. Here, the repetitive spikes in $D_\alpha$ emission indicate the ionization of neutral gas in the divertor region due to type-I ELM crashes, and a giant spike of $D_\alpha$ and line-averaged-density ($\bar{n}_e$) indicates a plasma disruption due to core LM. The successful control of core and edge instabilities is indicated by the absence of spikes in $D_\alpha$ emission throughout the whole discharge period without disruption. Unlike the optimized $n = 1$ EFC case, the other EFCs have a core or edge instability, as shown in Fig. 2a. With the minimized core-EFC, plasma can operate without disruption. However, ELMs are not controlled in H-mode from 3 s to the end of the discharge, which would need additional control to suppress ELMs for safe tokamak operation. For an edge-resonant EFC case, the plasma disrupted early in the low-density phase at around 3 s due to a core LM. The three discharges operate with the same condition with plasma current ($I_p$) of 0.51 MA, toroidal field ($B_T$) of 1.8 T, neutral beam injected (NBI) power of 1.1 MW at the low-density phase, and 3.17 MW at the stationary phase. Standard RMP[11,15,42] for a similar discharge is also shown in Fig. 2, which leads to a disruption in L-mode.

### Balancing core and edge stabilization requirements

The optimized EFC (#26027) is well resonating with the edge plasma but it corresponds to an unusual operating point in the 3D coil space as compared with the standard RMP. This correction is determined based on the two most important resonant modes; edge-localized and edge-dominant resonant modes at the plasma boundary, as shown in Fig. 2b (see Methods). The optimized EFC in Fig. 2c has an increased edge resonant field slightly above the ELM suppression threshold (see Methods) from the minimized core-EFC case to compensate for the limitation of the coil current limit of 5 kA (see Methods). Note that edge-resonant EFC and standard RMP are not a suitable EFC approach as it fails to avoid the extensive overlap between the core and edge resonant response.

The optimized 3D path of the 3D coil phase space is also validated with a slow change in the 3D spectrum, unlike the previous case that

applies a fixed 3D spectrum to control core and edge instabilities throughout the entire discharge period. The target plasma operates at the more elongated shaping ($\kappa \sim 1.86$ VS 1.73) but at the same $B_T$ of 1.8 T and similar $q_{95} \sim 5.1$. At 5 s, EFC is designed to localize the edge RMP with nearly zero $\delta B_{core}/\delta B_{edge}$ as shown in Fig. 3. The EFC reduces most of the core RMP from the proxy error field ($I_T = 5$ kA), but it maintains a finite edge RMP for ELM control. This EFC does not drive a core LM, but it leads to confinement degradation and ELM mitigation but not ELM suppression, as shown in Fig. 3a. Then, the EFC changes in time to increase the edge RMP by relaxing the core resonant field constraints from 5 s to 10 s. Note that 3D coil constraints keep the minimum level of $\delta B_{core}/\delta B_{edge}$ at a given $\delta B_{edge}$. This change of EFC gradually degrades stored energy, suppresses ELMs at 7.1 s, and further degrades the stored energy at the ELM-suppressed phase until 10 s, as shown in Fig. 3a. This dynamic 3D path can also significantly improve the conventional empirical approach of RMP ELM suppression that increases coil currents at fixed 3D spectrum. Optimization can take the most efficient 3D coil phase space to increase edge RMP, benefiting from the flexibility of the 3D coil phase space in KSTAR. This approach will be more useful in ITER as it has greater flexibility in 3D coil phase space than any existing tokamak.

### Optimizing confinement within the stable operation space

The improved confinement due to edge localization is also observed by ramping up the fixed spectrum of the 3D field to get the ELM-suppressed H-mode. Figure 4 compares experiments with coil current ramps of the different 3D spectrum, and their edge normalized resonant field profiles are shown in Fig. 4b. In each case, ELMs are strongly mitigated and then suppressed, and the ELM suppression lasts until the coil currents become maximum or large enough to cause a core LM disruption. Figure 4c, d, f, g compare the density measured with Thomson scattering[43] and plasma rotation profiles measured with charge exchange spectroscopy (CES)[44] before each 3D field and after the ELM suppression. Before the application of the 3D field, the initial

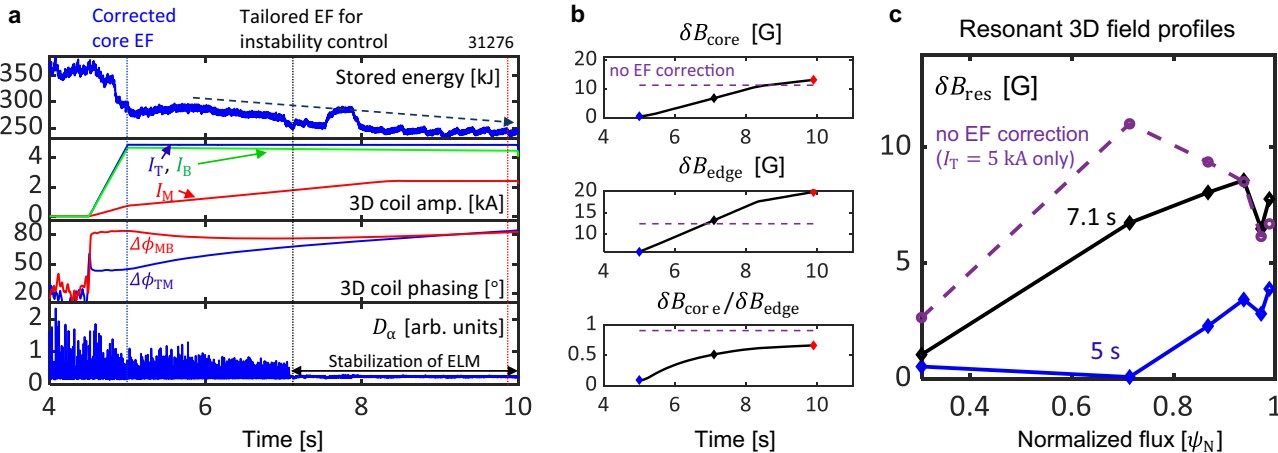

**Fig. 3 | Optimized path to edge stability maintaining core stability. a** From top to bottom: time evolution of the plasma stored energy showing confinement degradation, dynamic change of 3D coil variables ($I = (I_T, I_M, I_B, \phi_{MT}, \phi_{BM})$), and $D_\alpha$ emission (particle recycling light) showing ELM crashes and ELM suppression. **b** From top to bottom: time traces of the core resonant field ($\delta B_{core}$), and edge resonant field ($\delta B_{edge}$), and their ratio ($\delta B_{core}/\delta B_{edge}$) due to the optimal change of 3D coil variables. **c** The comparison of resonant field profiles ($\delta B_{res}$) at 5 s (blue), at 7.1 s (black) with the optimized error field correction, and with the proxy error field $I_T = 5$ kA without the error field correction (violet, dashed).

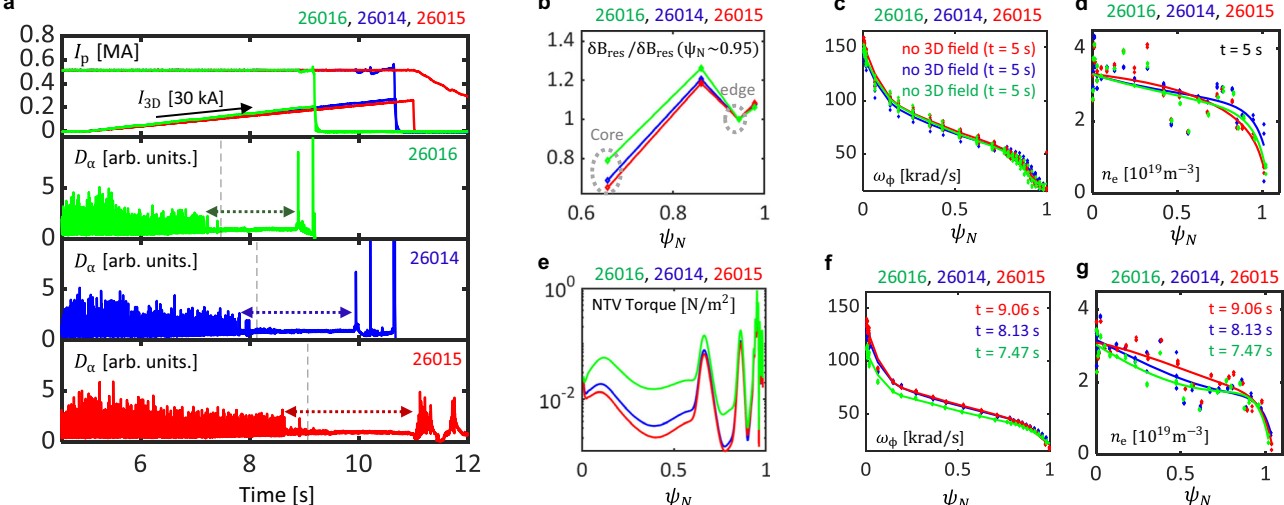

**Fig. 4 | Comparison of transport with different 3D field spectra. a** Time traces of plasma current ($I_p$), root sum squared coil current amplitude ($I_{3D}$), and $D_\alpha$ emission with different 3D field spectra. **b** edge normalized resonant field profiles ($\delta B_{res}/\delta B_{res}(\psi_N \sim 0.95)$) with a different 3D field spectrum. **c, d** Toroidal rotation frequency ($\omega_\phi$) and density ($n_e$) profiles before 3D fields at $t = 5.00s \pm 0.02s$. **e** Calculated Neoclassical Toroidal Viscosity (NTV) torque profiles. **f, g** Toroidal rotation frequency and density profiles after ELM suppression at $t = 9.06s \pm 0.02s$, $t = 8.13 \pm 0.02s$, and $t = 7.47 \pm 0.02s$ of discharge, #26016 (green), #26014 (blue), #26015 (red), respectively. The selected time points are indicated with a vertical dashed line in **a**.

density and rotation profiles are almost identical for the three cases indicating that discharges are well reproduced. On the other hand, density and rotation degradation in three cases have different degradation after the onset of ELM suppression. The reduced core-degradation of rotation in #26015 (red) compared to #26016 (green) is consistent with a reduction in core-resonant response. Additional neoclassical toroidal viscosity (NTV)[45] torque calculations using GPEC[46] also shows that the reduction of NTV in #26015 (red) as shown in Fig. 4e, which is consistent with reduced degradation of rotation. The limited improvement of plasma confinement, in this case, can be explained by engineering constraints of 3D coils, which leaves the residual core resonant field (Fig. 4b) and NTV torque (Fig. 4e) even with this optimization. This implies that a physics-based 3D coil design based on the edge localization scheme[28] can further improve plasma confinement by further reducing these core components. Note that

this ERMP example is one of many examples of its safe and efficient ELM suppression in KSTAR. For example, ERMP has improved the plasma confinement of other KSTAR discharges to extend its $\beta_N$ boundary up to $\beta_N$ of 2.65[47]. ERMP also demonstrated $n = 1$ RMP ELM suppression at ITER relevant $q_{95}$ of 3.5 in KSTAR[48].

## Control of edge transport barrier formation

Optimization of EFC also controls the edge transport barrier (ETB) formation and H-mode transition, triggered by a zonal flow and $E \times B$ shear[49–51] in tokamaks, while maintaining the $n = 1$ field at low density[52] without locking and disruption. Figure 5a, d compare the $n = 1$ EFC to access and to avoid the H-mode transition. Figure 5a shows more standard EFC to access H-mode, which shows ETB formation at $t \approx 2.47s$ with NBI power of 1.1 MW (#26026), where the reduction of coherence measured with electron cyclotron emission imaging (ECEI), and rapid

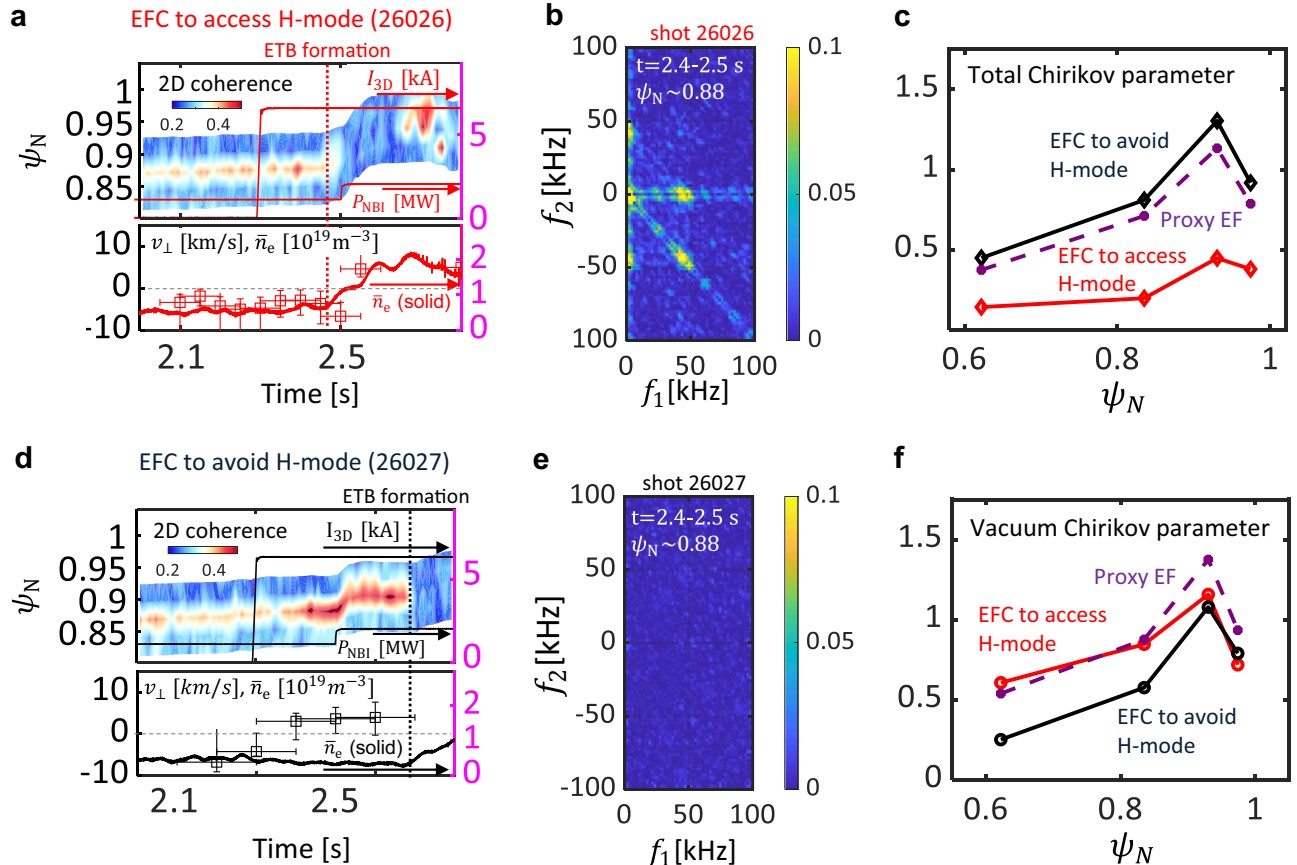

**Fig. 5 | Avoidance of ETB formation by optimizing error field correction.**
**a**, **d** (Top) Time evolution coherence measured with electron cyclotron emission imaging (ECEI), NBI power ($P_{NBI}$), root sum squared coil current ($I_{3D}$), and (bottom) perpendicular mode rotation ($v_\perp$) measured with ECEI at the maximum coherence, and edge line averaged density ($\bar{n}_e$) with error field correction (EFC) to access H-mode (**a** #26026) and to avoid H-mode (**d** #26027). In order to obtain $v_\perp$, the cross-phase was calculated using two channels in the vertical direction, and the speed was calculated using the slope of the coherent phase. The x-axis error bars in $v_\perp$ represent the time window of data for cross-phase calculation, and the y-axis error bar in $v_\perp$ represents 95% confidence limits for linear fitting. **b**, **e** Conditionally ensemble-averaged auto-bicoherence of the density fluctuations before the additional heating power with EFC to access H-mode (**b** #26026), and to avoid H-mode (**e** #26027). **c**, **f** Comparison of the Chirikov parameters of the island established with (**c**, total Chirikov) and without (**f**, vacuum Chirikov) 3D plasma response with EFC to access H-mode (red), to avoid H-mode (black), and proxy error field $I_T = 5\,kA$ without error field correction (violet, dashed).

increase of edge density indicates the formation of an ETB. At 2.47 *s*, limit cycle oscillations and particle and electron energy confinement enhancement are initiated during the slow change of plasma shape without additional heating power. This result is similar to the observation of zonal flow oscillations in DIII-D[53]. During this period, the squared bicoherence of density fluctuations ($\tilde{n}$) (see Methods) indicates a nonlinear three-wave coupling of high frequency turbulence and a low frequency zonal flows as shown by the $f_1 = -f_2$ line in Fig. 5b. The measurement implies the presence of a nonlinear interaction of turbulence and low-frequency zonal flow[54] contributes to the ETB formation. For the EFC to avoid H-mode transition, on the other hand, the squared bicoherence does not show any clue of nonlinear three-wave coupling, as shown in Fig. 5e. These are consistent with the theoretical prediction of zonal flow reduction by the applied 3D field[55]. Instead, the applied 3D field increases the 2D coherence before the additional heating power applied at 2.5 *s*. The edge perpendicular mode rotation $v_{\perp,ped}$ from ECEI at the maximum coherence region and line averaged density are also slightly reduced as shown in Fig. 5d, which could imply a penetration of the edge 3D field. No such behavior is observed in #26026 and other discharges without the applied 3D field. The EFC to avoid H-mode (#26027) eventually starts ETB formation at $t \approx 2.69s$ with the 81% increased heating power of 2.0 *MW*. This result agrees with the increase of the L-H transition power threshold due to the applied 3D field[42]. In

addition, this result also highlights the importance of the 3D plasma response to the EFC in controlling the H-mode transition. As two EFCs have different poloidal spectra, the experiment can investigate the reliability of the predicted plasma response model, which exhibits a higher sensitivity to specific components or distributions of the applied 3D field. The EFC to avoid H-mode has a weaker vacuum resonant response than the EFC to access H-mode, unlike the total resonant response which includes the plasma response. This is indicated by the Chirikov parameter of the island overlap[56] in Fig. 5c, f, which represents the strength of the resonant field with plasma response (total Chirikov) and without plasma response (vacuum Chirikov). However, the H-mode avoidance is only observed for the EFC more aligned with the edge-dominant resonant mode and confirms the importance of the plasma response in controlling the ETB formation.

### Avoidance of H-mode transition to sustain advanced plasma regime
Furthermore, the $n = 1$ ERMP can robustly sustain improved core confinement by avoiding H-mode transition, as shown in Fig. 6. The target discharge is an I-mode scenario in KSTAR, and it has improved core confinement with a core ion temperature up to 10 $keV$[38]. However, the improved core confinement is difficult to sustain without an ERMP and vanishes with the further development of an ETB. A discharge

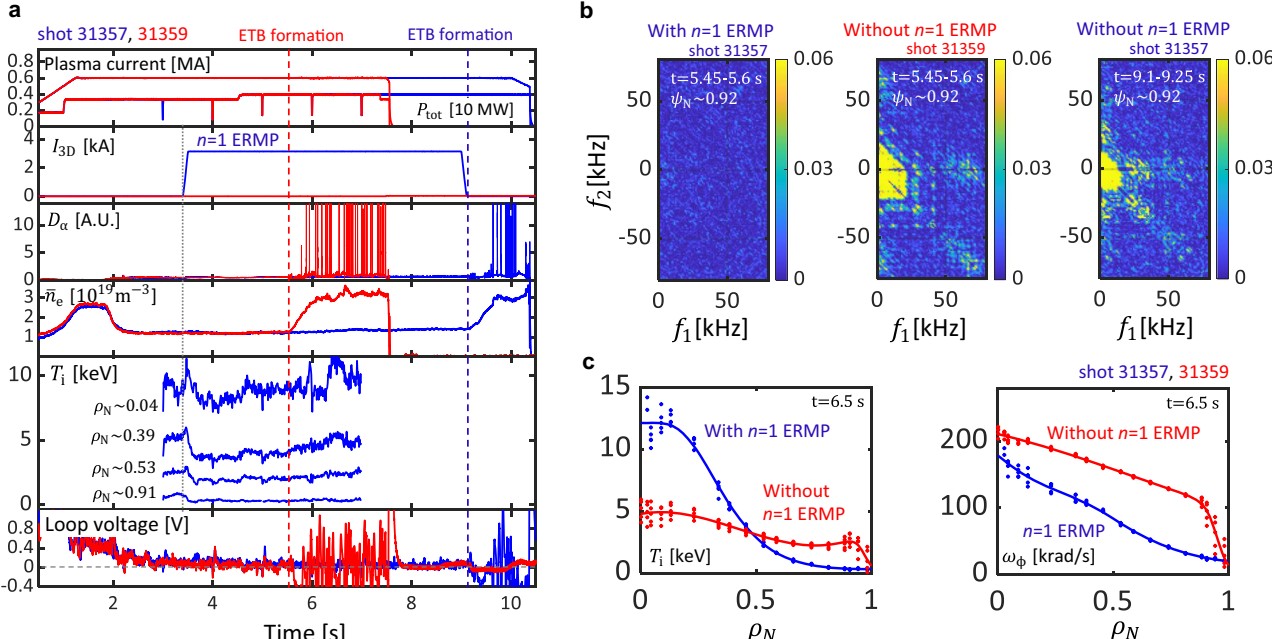

**Fig. 6 | Avoidance of an H-mode transition with an ERMP. a** From top to bottom: The time evolution of plasma current, total external heating power $P_{tot}$, root sum squared coil current ($I_{3D}$), $D_\alpha$ emission, line averaged density ($\bar{n}_e$), ion temperature ($T_i$) at different normalized toroidal minor radius defined by toroidal flux ($\rho_N$) and loop voltage. **b** Conditionally ensemble-averaged auto-bicoherence of the density fluctuations with and without $n=1$ ERMP. **c** The ion temperature $T_i$ (left) and toroidal rotation frequency $\omega_\Phi$ (right) with and without $n=1$ ERMP.

without an $n=1$ ERMP (shot 31359) begins to form an ETB in density at 5.5 s, as indicated by the increase of $\bar{n}_e$, $\beta_N$ and $\tilde{n}$ bicoherence. The formation of the ETB then leads to ELMs as shown by the $D_\alpha$ spikes. After the H-mode transition, as shown in Fig. 6c, the core ion temperature falls to $T_i \sim 5\,keV$. This reduction of $T_i$ is particularly unfavorable for the cross-section for fusion reactions. On the other hand, the discharge with an $n=1$ ERMP sustains the improved core confinement until 9.1 s, where we intentionally turn off the $n=1$ ERMP at 9.1 s to see its effect. Without the $n=1$ ERMP, an ETB starts to develop from 9.1 s as indicated by the increase of $\bar{n}_e$ and $\beta_N$, with the increase of squared bicoherence $\tilde{n}$. Note that the discharge operates with $n=1$ fields at a density of $\bar{n}_e \sim 1.2 \times 10^{19}\,m^{-3}$ without locking. There is a reduction of core ion temperature when the $n=1$ ERMP is applied at 3.4 s, but it remains at around 9 keV and then recovers to 10 keV by 6.5 s, as shown in Fig. 6a, c. These results show that H-mode avoidance using an $n=1$ ERMP not only eliminates the uncertainties in ELM destabilization but also helps long and robust sustainment of improved core confinement with $T_i \approx 100$ million kelvin by maintaining high fast particle fraction[38] at an almost fully non-inductive current drive fraction.

Although a tokamak is initially designed to be axisymmetric, these examples show that tokamaks can leverage the EFs to improve their stability and transport. The proposed method proves its robustness in various scenarios for correcting the most disruptive $n=1$ 3D field and highlights its potential for use in future reactors by combining EF and RMP. Its successful validation implies that ERMP can give insight into the design or upgrade of 3D coils to maximize its benefit[28]. Furthermore, the underlying theoretical framework is highly adaptable and can be extended to address challenges in burning plasmas, including the activity of Alfven eigenmodes induced by fusion products[57,58]. Its application to the torque matrix[11] can also improve plasma rotation control, which is essential in controlling various plasma instabilities to sustain a more favorable plasma regime in the fusion reactor. While the number of possible scenarios in the conventional post-evaluating approach grows exponentially with each additional 3D coil row, the ERMP approach will find the most efficient way to optimize the six rows of 3D coils in ITER for safer ELM control by combining EF and RMP. The

successful ERMP approach also inspires the design of future tokamaks with optimized 3D magnetic fields and coil geometries for their optimal plasma stability and confinement. The physics-based control method proposed in this paper also provides a foundation for the development of machine learning-based control, which will be presented in a future publication. These advancements in tokamak design and operation have the potential to contribute significantly to opening a path for the development of sustainable energy generation for the future.

## Methods

### Calculation of edge-dominant and edge-localized resonant modes

The edge-dominant and edge-localized resonant modes are calculated based on the ideal perturbed equilibrium code (IPEC)[59] and singular value decomposition (SVD) for a given equilibrium and a set of rational surfaces. In the ideal MHD perturbed equilibrium, the plasma responds to suppress the formation of the magnetic islands by the applied 3D fields and prevents the change of magnetic field topology. This leads to localized parallel shielding currents at the resonant layer, and the resonant field $\delta\mathbf{B}_{res}$ in the IPEC is the effective field that the shielding current would provide to offer the complete screening of the total resonant field. Therefore, this approach can include a global ideal MHD plasma response due to external 3D coil currents and calculates the total resonant field $\delta\mathbf{B}_{res}$ produced by the external coil current and the plasma response[23]. The approach finds the coupling matrix $\vec{\mathcal{C}}$ between the total resonant field $\delta\mathbf{B}_{res}$, and the externally applied field at the plasma boundary $\mathbf{V}_b^x$ by $\delta\mathbf{B}_{res} = \vec{\mathcal{C}} \cdot \mathbf{V}_b^x$, where $\mathbf{V}_b^x$ is the external field vector of poloidal Fourier harmonics.

Then the SVD of $\vec{\mathcal{C}}_{edge}$ provides an orthonormal basis of all the possible external fields that drive edge resonances that are sorted by the amplitude of that edge resonant drive. The edge-dominant resonant mode is given from the first right singular vector of the edge resonant coupling matrix, and it usually has a singular value much larger than the others and dominates the edge resonant drive. Here, $\vec{\mathcal{C}}_{edge}$ is chosen to couple the external field to the edge resonant field at

$0.9 < \psi_N < 0.99$. This is good enough to cover the pedestal of typical KSTAR H-mode plasmas, which is located somewhere between $0.9 < \psi_N < 0.99$. Note that the edge-dominant resonant mode drives core resonant fields in a tokamak due to the overlap of the core and edge resonant response.

The edge-localized resonant mode is calculated similarly but is designed to drive the edge resonant field without any core resonant field at $0 < \psi_N < 0.9$. This can be given by the edge-localized resonant coupling matrix $\vec{\mathcal{C}}_{e,cnull} = \vec{\mathcal{C}}_{edge} \cdot \vec{\mathcal{P}}_{c,null}$, where $\vec{\mathcal{P}}_{c,null}$ is a projection matrix to the core resonant null space[28]. In principle, this calculation can take any other metrics such as the NTV torque response matrix[46] or plasma response calculations from other 3D MHD codes.

### Calculation of ELM suppression threshold

The ELM suppression threshold is measured in KSTAR discharge 26014, shown in Fig. 4a, by gradually increasing coil currents with standard $n = 1$ RMP[11,15,42] ($I_{90°} = I_T/\sqrt{2} = I_M/\sqrt{2} = I_B/\sqrt{2}$, $\phi_{MB} = \phi_{MT} = 90°$). The critical coil current amplitude at ELM suppression ($I_{90°} = 1.8\,kA$) is used to estimate the edge resonant field threshold $\delta B_{edge,th}$. The resonant field at rational surfaces is calculated by IPEC, using an ideal MHD plasma response to represent the outer region of the resonant layer. The rational surfaces within $0.9 < \psi_N < 0.99$ are chosen to evaluate the edge resonant field threshold for ELM suppression, considering that the pedestal of KSTAR H-mode plasmas is located within this range. Although IPEC cannot describe the complex (non-ideal) inner layer dynamics, the narrowness of the resonant layers allows for a unified physics description of the inner and outer layer regions with asymptotic matching theory[20,21]. In the EFC experiments, the edge resonant fields are designed to stay above this threshold, $\delta B_{edge,th}$.

In addition, a nonlinear TM1 simulation[19] using experimentally measured profiles is conducted to investigate the physics mechanism of the ELM suppression threshold ($I_{90°} \sim 1.8$ kA). The calculated penetration threshold for at $q = 5$ rational surface in TM1 was $I_{90°} \sim 1.9$ kA with 10 % uncertainties, showing quantitative agreement with the measured empirical threshold. Considering $q_{95} \sim 5.1$ of the plasmas, this result implies that ELM suppression in these experiments is due to the penetration of the magnetic island at the top region of the pedestal.

### Increasing edge resonant field with 3D coil constraints

The edge resonant field is increased by changing the 3D spectrum from an edge-localized to an edge-dominant resonant mode with given 3D coil constraints to minimize the core resonant field. The change is made by penalizing the core resonant coupling matrix to relax the constraint of perfect nulls of the core resonant coupling by

$$\mathbf{B}_{edge} = \vec{\mathcal{C}}_{edge}^C \cdot \left( \vec{I} - c_{opt} \vec{\mathcal{R}}_{core} \right) \cdot \mathbf{I}_{coil}, \tag{1}$$

where $\mathbf{I}_{coil}$ is the complex vector representing the coil current and phase for each coil, $\vec{\mathcal{C}}_{edge}^C$ is the edge resonant coil coupling matrix, and $c_{opt} \vec{\mathcal{R}}_{core}$ is the weighted ($c_{opt} = 0 \sim 1$) removal of dominant coupling to $\mathbf{B}_{core}$ which are typically the most disruptive components of the error field. Here, $\vec{\mathcal{P}}_{c,null} = \vec{I} - \vec{\mathcal{R}}_{core}$, so $c_{opt} = 1$ represents perfect nulls of the core resonant coupling, and $c_{opt} = 0$ represents no constraints for core resonant coupling. For #26027 in Fig. 2, the core resonant field constraints were relaxed to raise the edge resonant field above the ELM suppression thresholds, which is taken using a single empirical operating point from the reference discharge (#26014). Note that this approach can work even without any empirical threshold, as shown in another example (Fig. 3, #31276).

### Bicoherence of density fluctuation

Density fluctuations ($\bar{n}$) for the bispectral analysis are measured with beam emission spectroscopy (BES) in KSTAR. The bispectral analysis of the conditionally ensemble-averaged BES signal is commonly used to show the three-wave non-linear interactions, as studied in KSTAR[60]. Here, squared bicoherence of $\bar{n}$ is defined by

$$\hat{b}^2(f_1 f_2) = \left| \left\langle \bar{n}^*(f = f_1 + f_2)\bar{n}(f_1)\bar{n}(f_2) \right\rangle \right|^2 \Big/ \left\langle \left| \bar{n}^*(f = f_1 + f_2) \right|^2 \right\rangle \left\langle \left| \bar{n}(f_1)\bar{n}(f_2) \right|^2 \right\rangle, \tag{2}$$

where the angle brackets denote ensemble averaging during the analysis period.

## Data availability
Raw data were generated from the KSTAR team. The data supporting the findings of this work are available from the corresponding author upon request.

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

## Acknowledgements

This work was supported by the U.S. Department of Energy under contract number DE-AC02-09CH11466. The United States Government retains a non-exclusive, paid-up, irrevocable, world-wide license to publish or reproduce the published form of this manuscript, or allow others to do so, for United States Government purposes. This work was also supported by Ministry of Science and ICT under KFE R&D Program of 'KSTAR Experimental Collaboration and Fusion Plasma Research (KFE-EN2401-15)'. This work was supported by the National Research Foundation (NRF) grant No. RS-2023-00281272 funded through MSIT, and also by the New Faculty Startup Fund from Seoul National University. This work was supported by the National Research Foundation of Korea under Grant No. 2019R1F1A1057545, No. 2022R1F1A1073863. This research was supported by the National R&D Program through the National Research Foundation of Korea (NRF) funded by the Ministry of Science & ICT (NRF-2019R1A2C1010757).

## Author contributions

S.M.Y. and J.-K.P. conceived the original idea. S.M.Y. and Y.M.J. led the experimental validations. S.M.Y., J.-K.P., and N.C.L. contributed to the development of the optimization scheme. J.L. diagnosed the temperature fluctuation using ECE imaging. S.M.Y., J.-K.P., N.C.L., Q.H., and S.K.K. analyzed the experiment and simulation results. J.H.L. diagnosed the plasma electron and density profiles using TS. J.W.K. diagnosed density fluctuation using BES. W.H.K. and H.H.L. diagnosed the ion temperature using charge-exchange spectroscopy. Y.S.N., T.S.H., G.J.C., and J.A.S. contributed to the clarification of the issues and the presentation of the results. H.H.L., G.Y.P., and W.H.K. participated in the design of the experiment and its execution. All authors reviewed the manuscript.

## Competing interests

The authors declare no competing interests.
