## [Peer Review File · Nature Communications]

Tailoring tokamak error fields to control plasma instabilities and transportREVIEWER COMMENTS

Reviewer #1 (Remarks to the Author):

The novel aspect of this work is to deploy low order non-axisymmetric fields in a tokamak (with toroidal periodicity of 1) to control edge instabilities, while controlling core resonances to minimize drives for core instabilities. While various devices (COMPASS-D, JET, EAST, KSTAR) have previously demonstrated ELM control with $n=1$ periodicity fields, without core modes destabilized, the key insight is to be able to do this without braking and increasing drives for modes in the core. This requires state-of-the-art physics models to properly capture the plasma response, and the development and experimental validation of this dual region consideration is the key accomplishment in this work. This is important because it offers a potentially more-applicable solution for future fusion reactors to control ELMs and core instabilities simultaneously, with the former requiring generation of edge resonances while the latter requires suppression of core resonances.

However, the opening context of the paper “Contrary to conventional wisdom, we report that the EFs in a tokamak can be favorably used for controlling plasma instabilities, such as edge-localized modes (ELMs),” is misleading and should be changed. In this case the authors are using coils to ‘simulate’ an error field. This amounts to exactly the same approach deployed for ELM control as on other devices mentioned above with $n=1$ fields. Therefore, it is incorrect to report that EFs can be favorably used for control as the key *new* observation from this paper; it is also implied by the earlier work. To address this, simply remove the phrase “Contrary to conventional wisdom, we report that the” – the rest of the sentence then simply accurately captures the previous work, and the next sentence captures the novel aspect of this work.

The previous work on $n=1$ ELM control should be briefly discussed in the introduction and cited. This includes: Liang, Phys. Rev. Lett. 98, 265004, 2007, Y. Sun et al 2017 Nucl. Fusion 57 036007, S.K. Kim et al 2022 Nucl. Fusion 62 026043, Fielding, Proc. 28th EPS, Funchal 2001, ECA Vol 25A 1825. $N=2$ ELM control was also demonstrated by M.J. Lanctot et al 2013 Nucl. Fusion 53 083019. It should be noted that all of these papers displayed various impacts on the ELM behavior and suppression, without driving core modes (though windows of operation were narrow, with locked modes emerging when fields were increased too high).

Page 3: the authors describe a unique feature of KSTAR, with flexible $n=1$ fields and low intrinsic field, but in fact COMPASS-D had this property (with 10 rows of coil bars). This device similarly combined $n=1$ fields to avoid core resonances while driving edge resonances (see Fielding, Proc. 28th EPS, Funchal 2001, ECA Vol 25A 1825) though with only type III ELM control and inducement of ELMs in ELF-free regimes. This work applied quite large amplitude fields at currents but avoided locked modes due to partial cancellation of core resonances.

The rest of this paper is a very nice elucidation of the sophisticated techniques and physics methods deployed, with state-of-the-art models being validated by the various observed behaviors, demonstrating the key physics and models needed to ensure optimal ELM control and avoid deleterious core effects. This really is at the forefront of the state of the art in this field, and represent significant results and capability for fusion energy. And it gratifying to see many aspects of this interaction considered in the paper such as transport, H mode access and NTV rotation.

Reviewer #2 (Remarks to the Author):

report on 'Tailoring tokamak error fields to control plasma instabilities and Transport' by S.M. Yang et al.

This manuscript reported an optimization of 3D field spectrum for stability control based on linear plasma response modeled by the GPEC code and its several applications for control of ELM and H mode access considering the avoidance of disruption and reduction of confinement degradation in the KSTAR tokamak. For ELM suppression in discharges with prescribed error field by one of the three rows of coils, the key idea for the optimization is to reduce the ratio of core harmonic plasma response to the edge ones. The reduction of core plasma response may avoid core harmonic mode locking and hence keep the require RMP field strength for ELM suppression less than the disruption limit and reduce the core plasma confinement degradation. The three rows of coils give more freedom in this optimization. For the H mode access with 3D field studies, it was shown that the 3D field can increase the L-H transition threshold power. This was applied for the avoidance of H mode transition to sustain high core ion temperature operation. The overall studies presented here may be good for scenario development utilizing 3D magnetic perturbations in KSTAR. However, it is hard for me to find any novelties in the manuscript that meet the high standard requirements for publications in the journal Nature communications. The fundamental physics about linear plasma response and the optimization of 3D field spectrum for ELM control have been intensively studied in the last ten years by many authors on different machines including KSTAR [Ref 11 for example]. Three rows of coils are indeed favorable for this optimization; however, it was very limited which means a narrow window as also shown in Ref 11. The quantitatively minor improvement of plasma confinement shown in figure 4 (only density was shown rather than temperature or energy) looks be almost negligible compared to the significant drop of stored energy shown in figure 3. Therefore, its impact for fusion research seems also to be limited. There were also many reports on increase of L-H transition power threshold by 3D field. The sustain of high core ion temperature by avoid L-H transition using 3D field is only one of the applications of this study. The physics is just that the H-mode is unfavorable to access high core ion temperature operation in this case, rather than the 3D physics itself. Unfortunately, I cannot recommend the publication of this manuscript in the journal Nature communications.

Reviewer #3 (Remarks to the Author):

The manuscript deals with an important issue in fusion energy research based on magnetic confinement in a tokamak namely suppression of edge localized modes (ELMs) using external coils. The ELM suppression using dedicated resonant magnetic perturbation (RMP) coils is already an old idea but this manuscript investigates combining the RMP coils with the error fields (EF) and error field correction coils (EFCC) to affect the ELMs and edge transport barrier formation. The manuscript reports on the experiments on the KSTAR tokamak where the ELM suppression was achieved and the associated modelling. If the method can be implemented in future tokamak based fusion reactors, this is a major breakthrough.

While the experimental result is exciting the description of it and the physical mechanisms and theoretical explanation behind the ELM suppression is not very clearly described in the paper. The ELM suppression threshold is mentioned (line 107) but it is unclear how that threshold was determined. This part of the manuscript should be made much clearer. Without a clearer description of the ELM suppression mechanism it is difficult to determine how relevant the optimization method described in the paper is to the actual suppression of ELMs.

In addition to that the following smaller questions and comments should be addressed before I can recommend the paper to be published in Nature Communications:

Line 34: The cancelling of the $n=1$ error field due to misalignment is not related to the $n=3$ or $n=4$ RMPs produced by the ELM suppression coils. So, I don't think it's really "contrary" to the efforts to cancel the EF rather than a completely separate issue.

Figure 1a. What is the scale in the colour scheme for the perturbation? Does 1a really show the displacement of the flux surfaces (how I would interpret the term "perturbed flux surface") or the perturbation of the magnetic field as in 1b?

Figure 1b: What is the definition of δB_n ?

Line 49. Instead of nuclear contamination you must be thinking of nuclear degradation of the coils.

Line 80. The term resonant should be defined. It is used a lot in the rest of the paper. What is resonating with what?

Line 90. D_α emission does not indicate the hard sputtering of deuterium ions, but is due to ionization of neutral gas in the divertor region.

107. This relates to my general comment. The threshold is mentioned here the first time. It should be defined and in particular it should be explained how the threshold is used in the following modelling.

Figure 4. The letters and the caption descriptions do not match. The caption description does not even say anything about f and g plots. Also d is obviously not “NTV torque profile” but most likely density profile.

Last paragraph of page 8: Is there an explanation why the plasma response increases the Chirikov parameter in one case and decreases it in the other?

Line 189-190: What sustains higher ion temperature gradient in the ERMP case?

RESPONSES TO REVIEWER #1's COMMENTS:

We authors thank the reviewer for providing important questions and comments, which have been very helpful for us to clarify the issues and improve the presentations of our work. Accordingly, we made modifications throughout the paper, reflecting our responses to the reviewer as well as the other reviewers. Please see the highlighted (blue) changes in a separate PDF file attached to this resubmission package and our responses in detail to each of the reviewer's comments below.

General Comment:

The novel aspect of this work is to deploy low order non-axisymmetric fields in a tokamak (with toroidal periodicity of 1) to control edge instabilities, while controlling core resonances to minimize drives for core instabilities. While various devices (COMPASS-D, JET, EAST, KSTAR) have previously demonstrated ELM control with $n=1$ periodicity fields, without core modes destabilized, the key insight is to be able to do this without braking and increasing drives for modes in the core. This requires state-of-the-art physics models to properly capture the plasma response, and the development and experimental validation of this dual region consideration is the key accomplishment in this work. This is important because it offers a potentially more-applicable solution for future fusion reactors to control ELMs and core instabilities simultaneously, with the former requiring generation of edge resonances while the latter requires suppression of core resonances.

Comment 1:

*However, the opening context of the paper "Contrary to conventional wisdom, we report that the EFs in a tokamak can be favorably used for controlling plasma instabilities, such as edge-localized modes (ELMs)," is misleading and should be changed. In this case the authors are using coils to 'simulate' an error field. This amounts to exactly the same approach deployed for ELM control as on other devices mentioned above with $n=1$ fields. Therefore, it is incorrect to report that EFs can be favorably used for control as the key *new* observation from this paper; it is also implied by the earlier work. To address this, simply remove the phrase "Contrary to conventional wisdom, we report that the" – the rest of the sentence then simply accurately captures the previous work, and the next sentence captures the novel aspect of this work.*

Response to comment 1:

Thank you, and we agree with the reviewer's comments. We removed the "Contrary to conventional wisdom, we report that the" to avoid those misinterpretations. Please see the highlighted changes in lines 15-16 as shown below.

Lines 15-16: "Meanwhile, the EFs in the tokamak can be favorably used for controlling plasma instabilities, such as edge-localized modes (ELMs), while maintaining a hot fusion plasma at a temperature of 100 million kelvin."

Comment 2:

The previous work on $n=1$ ELM control should be briefly discussed in the introduction and cited. This includes: Liang, Phys. Rev. Lett. 98, 265004, 2007, Y. Sun et al 2017 Nucl. Fusion 57 036007, S.K. Kim et al 2022 Nucl. Fusion 62 026043, Fielding, Proc. 28th EPS, Funchal 2001, ECA Vol 25A 1825. $N=2$ ELM control was also demonstrated by M.J. Lanctot et al 2013 Nucl. Fusion 53 083019. It should be noted that all of these papers displayed various impacts on the ELM behavior and suppression, without driving core modes (though windows of operation were narrow, with locked modes emerging when fields were increased too high).

Response to comment 2:

Thank you for the suggestion. We agree with this and cited those previous works and added a brief discussion about previous work on $n=1$ and $n=2$ ELM control. We revised the manuscript as shown below.

Lines 51-53: “For example, stable ELM control using $n = 1$ [33, 30, 31, 32, 15, 11] and $n = 2$ [34] RMPs typically requires operation within narrow windows, as an increase in the low- n RMP strength also results in the emergence of locked modes in the core.”

Comment 3.

Page 3: the authors describe a unique feature of KSTAR, with flexible $n=1$ fields and low intrinsic field, but in fact COMPASS-D had this property (with 10 rows of coil bars). This device similarly combined $n=1$ fields to avoid core resonances while driving edge resonances (see Fielding, Proc. 28th EPS, Funchal 2001, ECA Vol 25A 1825) though with only type III ELM control and inducement of ELMs in ELM-free regimes. This work applied quite large amplitude fields at currents but avoided locked modes due to partial cancellation of core resonances.

Response to comment 3:

We appreciate the reviewer's expertise in raising the concern and providing the relevant reference. We agree with this concern and remove the “unique.” in the modified manuscript. Please see the highlighted changes in lines 70-71, as shown below.

Lines 70-71: KSTAR has an advantage in investigating the design of the EFC over other tokamaks due to its low level of intrinsic $n = 1$ EF [41] and flexible three rows of the external 3D coils (Fig. 1a, 1b)

Comment 4.

The rest of this paper is a very nice elucidation of the sophisticated techniques and physics methods deployed, with state-of-the-art models being validated by the various observed behaviors, demonstrating the key physics and models needed to ensure optimal ELM control and avoid deleterious core effects. This really is at the forefront of the state of the art in this field, and

represent significant results and capability for fusion energy. And it is gratifying to see many aspects of this interaction considered in the paper such as transport, H mode access and NTV rotation.

Response to comment 4:

We authors are grateful to have the reviewer's supporting and helpful comments. We appreciate the reviewer's expertise, which summarizes the strength of this work even more efficiently than we did, and for opportunities to improve the clarity of our work from these valuable comments.

RESPONSES TO REVIEWER #2's COMMENTS:

We authors would like to express our gratitude to the reviewer for their valuable questions and comments, which have been very helpful for us to clarify the issues and improve the presentations of our work. Accordingly, we made various modifications throughout the paper, reflecting our responses to the reviewer as well as the other reviewers. Please see the highlighted (purple) changes in a separate PDF file attached to this resubmission package and our responses in detail to each of the reviewer's comments below.

General Comment:

This manuscript reported an optimization of 3D field spectrum for stability control based on linear plasma response modeled by the GPEC code and its several applications for control of ELM and H mode access considering the avoidance of disruption and reduction of confinement degradation in the KSTAR tokamak. For ELM suppression in discharges with prescribed error field by one of the three rows of coils, the key idea for the optimization is to reduce the ratio of core harmonic plasma response to the edge ones. The reduction of core plasma response may avoid core harmonic mode locking and hence keep the require RMP field strength for ELM suppression less than the disruption limit and reduce the core plasma confinement degradation. The three rows of coils give more freedom in this optimization. For the H mode access with 3D field studies, it was shown that the 3D field can increase the L-H transition threshold power. This was applied for the avoidance of H mode transition to sustain high core ion temperature operation. The overall studies presented here may be good for scenario development utilizing 3D magnetic perturbations in KSTAR. However, it is hard for me to find any novelties in the manuscript that meet the high standard requirements for publications in the journal Nature communications.

Response to general comment:

We appreciate the reviewer's comments that give us another opportunity to go over this important question. We acknowledge that the presentations of this work did not sufficiently highlight its novelties. Here, we want to emphasize that the novelty of this work is the systematic edge localization scheme and its first experimental validation with various applications. This dual region consideration is critical for effectively leveraging the error fields (EF) using error field correction coils (EFCCs) and RMP coils. These novelties and impacts are also acknowledged by the other two reviewers, as referred to below.

Reviewer #1: *"The novel aspect of this work is to deploy low order non-axisymmetric fields in a tokamak (with toroidal periodicity of 1) to control edge instabilities, while controlling core resonances to minimize drives for core instabilities. ... This requires state-of-the-art physics models ... This really is at the forefront of the state of the art in this field, and represent significant results and capability for fusion energy"*

Reviewer #3: *"The ELM suppression using RMP coils is already an old idea but this manuscript investigates combining the RMP coils with the error fields (EF) and error field correction coils (EFCC) to affect the ELMs and edge transport barrier formation. ... If the method can be implemented in future tokamak based fusion reactors, this is a major breakthrough."*

Without this systematic scheme, the dual region consideration is not straightforward in tokamaks due to the plasma response. To emphasize this point, we revised the manuscript as shown below (Please also see more detailed explanations in “**Response to Comment 2**”).

Lines 42-44: “What is missing in the standard EFC is the ability to isolate the edge from the core 3D effects, **which is challenging due to the strong poloidal coupling and kink mode amplification**”

Lines 56-57: “This ERMP is systematically given by characterizing core and edge resonant responses as coupled weakly damped oscillators.”

We also want to highlight the demonstrated safety of this tailored $n=1$ field at **low-density plasmas**¹, in addition to its utilization for ELM suppression and H-mode avoidance. To better highlight these points, we modified the manuscript as shown below.

Lines 155-157: “Optimization of EFC also controls the edge transport barrier (ETB) formation and H-mode transition, triggered by a zonal flow and $E \times B$ shear [49, 50, 51] in tokamaks, **while maintaining the $n = 1$ field at low density [52] without locking and disruption.**”

Lines 196-197: “Note that the discharge operates with $n = 1$ fields at a density of $n_e \sim 1.2 \times 10^{19} \text{m}^{-3}$ without locking.”

Furthermore, combining the RMP and the error field provides valuable insights for future reactor design and operation, as pointed out by reviewers #1 and #3 (Note that the most probable and disruptive EFs are low-order ($n=1$) non-axisymmetric fields). We also emphasize that EF correction is an essential topic in ITER as much as RMP. For example, ITER plans to have three rows of EFCCs in addition to the three rows of RMP coils, as shown in Fig. A. Also, there is a dedicated ITPA activity focused on error fields, i.e., MDC-19. A primary focus of this activity is disruption avoidance due to error fields, and tokamaks have spent extensive resources to eliminate these error fields. This work introduces a novel perspective largely overlooked in tokamak fusion plasma research, where the focus has traditionally been on eliminating error fields. Instead, this work proposes a method to leverage the error field to control plasma instability and transport. The demonstrated robustness of this new scheme, supported by multiple experimental validations, can substantially enhance the connection between the error field and RMP, offering new insight into the future reactor design and operation. To emphasize this point, we modified the manuscript as shown below.

Lines 205-207: “The proposed method proves its robustness in various scenarios for correcting the most disruptive $n = 1$ 3D field and highlights its potential for use in future reactors **by combining EF and RMP**”

Lines 212-215: “While the number of possible scenarios in the conventional post-evaluating approach grows exponentially with each additional 3D coil row, the ERMP approach will find the most efficient way to optimize the six rows of 3D coils in ITER for safer ELM control **by combining EF and RMP.**”

¹ $n=1$ 3D field typically locks and disrupts low-density plasmas, as reported in DIII-D, JET, COMPASS-U, ALCATOR C-mod, MAST, NSTX, JTEXT, EAST, and KSTAR [Scoville et al., NF 1991], [Buttery et al., NF 1999], [Buttery et al., NF 2000], [Wolfe et al., NF 2000], [Howell et al., 2007], [Menard et al., NF 2010], [Wang et al., NF 2014], [Logan et al., NF 2020], [Yang et al., NF 2021], etc.

Please also see our responses to other comments to reflect these points.

Figure A. Non-axisymmetric coil configuration planned for ITER. Reproduced/reprinted with permission from [D.B. Weisberg et al 2019 Nucl. Fusion 59 086060]

Comment 2:

The fundamental physics about linear plasma response and the optimization of 3D field spectrum for ELM control have been intensively studied in the last ten years by many authors on different machines including KSTAR [Ref 11 for example].

Response to comment 2:

We agree that linear plasma response has been intensively studied in the last ten years, including Ref. 11. We appreciate the reviewer's feedback and the opportunity to highlight the distinctions from previous studies. First, we would like to emphasize that a systematic scheme to decouple edge and core resonances effectively has not been developed in those previous studies (including Ref. 11). Unlike those previous works, the edge localization scheme characterizes core and edge resonant responses as coupled weakly damped oscillators. This can find the unique and most efficient 3D field that resonates with the edge and not the core. This is essential for effectively leveraging tokamak error fields as it finds a unique error field correction from unlimited possible choices of spatial spectra. Note that the challenge of decoupling the edge and core resonances is far from simple, even within the framework of linear plasma response, due to the strong poloidal mode coupling and resonant field amplification in tokamaks [A. Boozer, PRL 2001], [D. A. Ryan et al., PPCF 2015]. In this work, we present the first experimental validation of the null-space projection scheme to achieve systematic decoupling between core and edge resonant fields. As mentioned earlier, we revised the manuscript as shown below.

Lines 43-44: "What is missing in the standard EFC is the ability to isolate the edge from the core 3D effects, which is challenging due to the strong poloidal coupling and kink mode amplification."

Lines 56-57: "This ERMP is systematically given by characterizing core and edge resonant responses as coupled weakly damped oscillators."

In addition, this scheme can be easily extended to include other optimization metrics, contributing to a more comprehensive tokamak error field control. To better emphasize this point, we revised the manuscript as shown below.

Lines 210-212: “Its application to the torque matrix [11] can also improve plasma rotation control, which is essential in controlling various plasma instabilities to sustain a more favorable plasma regime in the fusion reactor.”

Another practical utility of this optimization is its potential to design more innovative EFC and RMP coils (refer to the response to comment 3). Its incorporation into the 3D coil design has a strong potential to improve its design without being constrained by a priori choices of the coil geometries. To emphasize this point, we revised the manuscript as shown below.

Lines 207-208: “Its successful validation implies that ERMP can give insight into the design or upgrade of 3D coils to maximize its benefit [28]”

Lines 215-216: “The successful ERMP approach also inspires the design of future tokamaks **with optimized 3D magnetic fields and coil geometries** for their optimal plasma stability and confinement.”.

Comment 3:

Three rows of coils are indeed favorable for this optimization; however, it was very limited which means a narrow window as also shown in Ref 11.

Response to comment 3:

Thank you for raising the concern regarding the limited window with three rows of coils in the previous study. It is precisely why we propose this scheme to take advantage of additional 3 rows of error field correction coils in ITER in addition to 3 rows of RMP coils (Fig. A), as pointed out by reviewer #3. The proposed scheme can efficiently control the 12-dimensional freedoms (6 amplitudes and 6 toroidal phases) in one toroidal mode number to increase the safe operation window. This will become important in ITER because the number of possible scenarios exponentially increases with each additional dimension. To better emphasize this point, we revised the manuscript as shown below.

Lines 212-215: “While the number of possible scenarios in the conventional post-evaluating approach grows exponentially with each additional 3D coil row, the **ERMP approach will find the most efficient way to optimize the six rows of 3D coils in ITER for safer ELM control by combining EF and RMP.**”

Also, the narrow window using existing KSTAR coils is another reason to design 3D coils based on the systematic edge localization scheme, as mentioned in response to comment 2. For example, edge localization provides insights into the optimal size and placement of 3D coils for ELM suppression. Figure B illustrates how modified coil location and size based on the edge localized RMP can increase the safe ELM suppression window. The windows are indicated by the expansion of the blue-shaded window (41 %) in Fig. B. Further improvement (141 %) is also possible based on the 3D geometry optimization of the coil

using the stellarator design tool for a given target field, as shown in Fig. B. To emphasize this point, we also revised the manuscript as shown below.

Lines 207-208: “Its successful validation implies that ERMP can give insight into the design or upgrade of 3D coils to maximize its benefit [28].

Lines 215-216: “The successful ERMP approach also inspires the design of future tokamaks **with optimized 3D magnetic fields and coil geometries** for their optimal plasma stability and confinement.”.

Figure B. (Top) Location of existing 3D coils and modified new coil rows with edge localized RMP. (Bottom) 3D coil phase space ($I_T = I_M = I_B, \phi$) predicted by $n = 1$ IPEC resonant field with the exiting coils and with the new coils modified with edge localized RMP. The polar plot's blue, red, and green regions represent 3D coil operating space with ELM suppression, disruption driven by LM, and weak resonance in KSTAR, respectively. Reproduced/reprinted with permission from [S.M. Yang et al. Localizing resonant magnetic perturbations for edge localized mode control in KSTAR. Nucl. Fusion 60 096023 (2020)]

Comment 4.

The quantitatively minor improvement of plasma confinement shown in figure 4 (only density was shown rather than temperature or energy) looks be almost negligible compared to the significant drop of stored energy shown in figure 3. Therefore, its impact for fusion research seems also to be limited.

Response to comment 4:

Thank you for sharing your concern about the minor improvement of plasma confinement. We also agree with the reviewer’s concern that confinement improvement is limited in this case. However, we have additional results supporting confinement improvement by edge localization as shown below.

1. Beta increase with edge localization in KSTAR experiment [M.W. Kim et al., NF 2023]

- The reduction of the core resonant field leads to about 0.1 increase in maximum β_N (Fig. C(c) vs. C(d)), reaching $\beta_N \sim 2.7$ (highest value in KSTAR) with RMP-ELM suppression. The integrated optimization changes the 3D coil currents with adaptive ELM control and edge localized RMP (ERMP, Fig. C(d)), and it is compared with the same approach using conventional RMP (CRMP, Fig. C(c)). This shows that minimizing the core resonant field with edge localization can also improve energy confinement (Fig. C(e)) and extend operational boundaries. **We included this paper as a new reference [47] (line 152, lines 316-319) in the modified manuscript.**

Figure C. Comparison of β_N during the $n = 1$ RMP-driven ELM crash control. (a) Total auxiliary heating power. Time traces of β_N , D_α (cyan), and RMP coil current (black) in (b) the CRMP with pre-set constant I_{RMP} (#31185), (c) the CRMP with adaptive feedback control (#31184), and (d) the ERMP with adaptive feedback control (#31189). (e) H-factor ($H_{89L} = \tau_{E,exp} / \tau_{E,89L}$) (f) Comparison of resonant magnetic field (δB_{res}) profiles with CRMP (#31184,#31185) and ERMP (#31189). [Minwoo Kim et al. Integrated RMP-based ELM-crash-control process for plasma performance enhancement during ELM crash suppression in KSTAR. Nucl. Fusion 63 086032 (2023)]

2. Fast ion confinement improvement by keeping KAM surface using edge localization [S.M. Yang et al., submitted to NF]
 - A numerical simulation of fast ion transport using NuBDec [Rhee et al., POP 2019] shows improved fast ion confinement with ERMP, as shown in Fig. D(b). The physics of this improvement is the presence of a good Kolmogorov–Arnold–Moser (KAM) surface that reduces the fast particle losses, as shown in Fig. D(c), D(d), and D(e). The results highlight the potential of the proposed edge localization scheme to improve the confinement of fast particles, which can carry significant energy in fusion reactors.

Figure D. The comparison of (a) resonant field profiles, (b) the toroidal average of the estimated fast ion loss and core resonant field at the 2/1 rational surface. Poincaré map of fast particles with three different resonant field profiles with (c) reference case, (d) 90-degree phasing, and (e) ERMP. Reproduced/reprinted with permission from [S.M. Yang et al. Tailoring resonant magnetic perturbation to optimize fast-ion confinement during ELM control in KSTAR. Nucl. Fusion 63 126046 (2023)]

We didn't include all these results in the manuscript to have the essential validation of the edge localization scheme with limited space in this paper. Meanwhile, we prepared other publications with the above results. Some of these results are led by another group (with collaboration), and we didn't have enough space to include all these results properly in this manuscript. We hope the reviewer can count these results as supporting evidence. To emphasize this point, we also modified the manuscript as shown below.

Lines 150-153: "Note that this ERMP example is one of many examples of its safe and efficient ELM suppression in KSTAR. For example, ERMP has improved the plasma confinement of other KSTAR discharges to extend its β_N boundary up to β_N of 2.65 [47]. ERMP also demonstrated $n = 1$ RMP ELM suppression at ITER relevant q_{95} of 3.5 for the first time in KSTAR [48]."

Furthermore, we would like to emphasize the potential of edge localization for further improvement in plasma confinement. Even with the implementation of edge localized RMP, as shown in Figure E, residual core resonant fields remain due to the engineering constraints (5kA maximum) of the 3D coils, as depicted in Figure E. However, Figure E also shows that by increasing the allowed coil current from 5 to 10 kA, the edge localization scheme can further reduce these core resonant fields. This finding suggests that additional enhancements of confinement can be achieved through a physics-based EFC or RMP coil design.

Based on these results, KSTAR has initiated an upgrade of the 3D coil configurations and power supplies to investigate further improvements in plasma confinement with ELM suppression. Specifically, the proposal targets the top and bottom rows of KSTAR's 3D coils, as these rows are crucial for enhancing edge localization. The upgrades of the 3D coil configurations and power supplies are expected to be completed before the upcoming KSTAR campaign, which aims to validate the additional improvement of plasma confinement with ELM suppression.

To emphasize these points, we modified the manuscript as shown below.

Lines 146-149: "The limited improvement of plasma confinement, in this case, can be explained by engineering constraints of 3D coils, which leaves the residual core resonant field (Fig. 4b) and NTV torque (Fig. 4e) even with this optimization. This implies that a physics-based 3D coil design based on the edge localization scheme [28] can further improve plasma confinement by further reducing these core components."

Figure E. The comparison of resonant field profiles from the discharges #26016, #26014, #26015 (ERMP with 5 kA constraint of 3D coil currents) and ERMP with a relaxed 10 kA constraint of 3D coil currents

Comment 5.

There were also many reports on increase of L-H transition power threshold by 3D field. The sustain of high core ion temperature by avoid L-H transition using 3D field is only one of the applications of this study. The physics is just that the H-mode is unfavorable to access high core ion temperature operation in this case, rather than the 3D physics itself.

Response to comment 5:

Thank you for this comment, and we agree that there were many reports on the increase of the L-H transition power threshold with the 3D field. Therefore, we tried to emphasize the uniqueness of this work, a more sophisticated optimization approach by considering both core and edge resonant fields in the modified manuscript. As mentioned in the previous responses (e.g., Response to general comment), this optimization is critical in utilizing the n=1 (error) field to increase the L-H transition power threshold while simultaneously avoiding its disruptive core response, particularly in low-density plasmas.

As investigated in previous studies, the n=1 3D field, which is the most probable error field in the tokamak, presents challenges in maintaining stability at low-density plasmas². Again, our objective is to highlight both the safety and efficiency of our approach, at these low densities (e.g. $\bar{n}_e \sim 1.2 \times 10^{19} m^{-3}$), which also demonstrates its practicality in sustaining high core ion temperature using the n=1 3D field. We tried to emphasize this point in the modified manuscript as shown below.

² [Scoville et al., NF 1991], [Buttery et al., NF 1999], [Buttery et al., NF 2000], [Wolfe et al., NF 2000], [Howell et al., 2007], [Menard et al., NF 2010], [Wang et al., NF 2014], [Logan et al., NF 2020], [Yang et al., NF 2021], etc.

Lines 155-157: “Optimization of EFC also controls the edge transport barrier (ETB) formation and H-mode transition, triggered by a zonal flow and $E \times B$ shear [49, 50, 51] in tokamaks, **while maintaining the $n = 1$ field at low density [52] without locking and disruption.**”

Lines 196-197: “Note that the discharge operates with $n = 1$ fields at a density of $n_e \sim 1.2 \times 10^{19} \text{m}^{-3}$ without locking.”

Another crucial aspect of this study is the role of plasma response in preventing the L-H transition. We conducted a comparison between cases with two different poloidal spectra, leading to contrasting outcomes regarding the Chirikov criteria of $n=1$ fields. This comparison serves as another validation of the edge localization scheme, contributing to a deeper understanding of the physics involved in 3D field optimization for the L-H transition. We tried to emphasize this point in the modified manuscript as shown below.

Lines 177-179: “As two EFCs have different poloidal spectra, the experiment can investigate the reliability of the predicted plasma response model, which exhibits a higher sensitivity to specific components or distributions of the applied 3D field.”

In summary, the proposed optimization schemes that control both core and edge resonant fields are critical to tailor the error field (low- n 3D field) to optimize various tokamak scenarios, as shown below.

1. FIRE-mode [Han et al., Nature 2023]: **Increased edge resonant field with minimized core resonant field.**
 - ⇒ **Core resonant field:** Minimized to avoid locked mode at low-density FIRE mode.
 - ⇒ **Edge resonant field:** Increased to avoid H-mode transition.

2. RMP-ELM suppressed H-mode: **Increased edge resonant field with minimized core resonant field.**
 - ⇒ **Core resonant field:** Minimized to improve the confinement of ELM-suppressed H-mode and to avoid locked mode.
 - ⇒ **Edge resonant field:** Increased to suppress ELMs.

In addition, the adaptability of the edge localization scheme can incorporate other metrics, such as the NTV (Neoclassical Toroidal Viscosity) torque matrix [Park et al., POP 2017], into the optimization process. Although not shown in this paper, this inclusion can enhance the development of other tokamak scenarios, such as the Quiescent H-mode (QH mode), as shown below.

3. QH mode: Increase edge non-resonant field with minimized core resonant field
 - ⇒ **Core resonant field:** Minimized to avoid locked mode.
 - ⇒ **Edge non-resonant field:** Increased to tailor edge rotation profile.

We tried to reflect these points in the outlook session of the modified manuscript.

Lines 210-212: “Its application to the torque matrix [11] can also improve plasma rotation control, which is essential in **controlling various plasma instabilities to sustain a more favorable plasma regime in the fusion reactor.**”

RESPONSES TO REVIEWER #3's COMMENTS:

We authors are grateful to have the reviewer's supporting and helpful comments and, thus, opportunities to improve the clarity in the presentation of our work. We accordingly made various modifications in this revision, as explained in detail below. Please check the highlighted (red) changes in a separate PDF in this resubmission package.

General comment 1)

The manuscript deals with an important issue in fusion energy research based on magnetic confinement in a tokamak namely suppression of edge localized modes (ELMs) using external coils. The ELM suppression using dedicated resonant magnetic perturbation (RMP) coils is already an old idea but this manuscript investigates combining the RMP coils with the error fields (EF) and error field correction coils (EFCC) to affect the ELMs and edge transport barrier formation. The manuscript reports on the experiments on the KSTAR tokamak where the ELM suppression was achieved and the associated modelling. If the method can be implemented in future tokamak based fusion reactors, this is a major breakthrough.

General comment 2)

While the experimental result is exciting the description of it and the physical mechanisms and theoretical explanation behind the ELM suppression is not very clearly described in the paper. The ELM suppression threshold is mentioned (line 107) but it is unclear how that threshold was determined. This part of the manuscript should be made much clearer. Without a clearer description of the ELM suppression mechanism it is difficult to determine how relevant the optimization method described in the paper is to the actual suppression of ELMs.

Response to general comment 2:

Thank you for asking about this important issue. The ELM suppression threshold is taken from the discharge 26014 shown in Fig. A(a) by slowly increasing the coil current at $n=1$, 90-degree phasing. This $n=1$, 90-degree phasing has been a standard RMP ([Jeon et al., PRL 2012], [Park et al., NP 2018], Fig. 2c in the manuscript) for ELM suppression in KSTAR. The critical coil current at the ELM suppression threshold, $I_{90\text{-deg}} \sim 3.6$ kAt (considering two coil turns), is used to estimate the edge resonant field threshold δB_{edge} , edge resonant field at $0.9 < \psi_N < 0.99$ that covers the pedestal top region. The H-mode phase of all EFC experiments is operated at the same condition of discharge 26014 (with the same plasma current, toroidal magnetic field, plasma shape, heating power, etc.).

Also, the physics mechanism of the ELM suppression threshold is understood as an edge island penetration at the pedestal top. This is supported by the estimated edge island penetration threshold by TM1 simulation [Q. Hu et al., PRL 2020] for $q=m/n=5/1$ island, as shown in Fig A. The 5/1 island threshold shows a good agreement with a measured empirical threshold of $I_{90\text{-deg}} \sim 3.6$ kAt, considering 10 % uncertainty comes from the transport coefficients.

Figure A. (a) The time traces of 3D coil current at 90-degree phasing (b) D_α showing ELM crashes and ELM suppression in the discharge 26014. (c) TM1 simulated edge island width at $n=1$, 90-degree phasing.

We have modified the manuscript to add a more detailed description as follows.

Lines 372-388: “The ELM suppression threshold is measured in KSTAR discharge 26014, as shown in Fig. 4a, by gradually increasing coil currents with a standard $n = 1$ RMP [15, 42, 11] ($I_{90^\circ} = I_T/\sqrt{2} = I_M/\sqrt{2} = I_B/\sqrt{2}$, $\phi_{MB} = \phi_{MT} = 90^\circ$). The critical coil current amplitude for ELM suppression ($I_{90^\circ} = 1.8$ kA) is used to estimate the edge resonant field threshold $\delta B_{\text{edge,th}}$. The resonant field at rational surfaces is calculated using IPEC, which employs an ideal MHD plasma response to represent the outer region of the resonant layer. The rational surfaces within $0.9 < \psi_N < 0.99$ are selected to evaluate the edge resonant field threshold for ELM suppression, given that the pedestal of KSTAR H-mode plasmas is situated within this range. Although IPEC cannot capture the complex (non-ideal) dynamics of the inner layer, the narrowness of the resonant layers permits a unified physics description of both the inner and outer layer regions through asymptotic matching theory [20, 21]. In the EFC experiments, the designed edge resonant fields are maintained above this threshold, $\delta B_{\text{edge,th}}$.

In addition, a nonlinear TM1 simulation [19] using experimentally measured profiles is conducted to investigate the physics mechanism of the ELM suppression threshold ($I_{90^\circ} \sim 1.8$ kA). The calculated penetration threshold for at $q = 5$ rational surface in TM1 was $I_{90^\circ} \sim 1.9$ kA with 10 % uncertainties, showing quantitative agreement with the measured empirical threshold. Considering $q_{95} \sim 5.1$ of the plasmas, this result implies that ELM suppression in these experiments is due to the penetration of the magnetic island at the top region of the pedestal.”

Comment 1)

In addition to that the following smaller questions and comments should be addressed before I can recommend the paper to be published in Nature Communications:

Line 34: The cancelling of the $n=1$ error field due to misalignment is not related to the $n=3$ or $n=4$ RMPs produced by the ELM suppression coils. So, I don't think it's really "contrary" to the efforts to cancel the EF rather than a completely separate issue.

Response to comment 1:

Thank you for your comment. We have modified the sentences with the phrase 'simultaneously with' based on the reviewer's comment. Please refer to line 33 in the modified manuscript.

Lines 33-35: "Simultaneously with extensive efforts to reduce the 3D EFs, ITER has adopted a small externally applied 3D field to prevent excessive material erosion [8, 9] due to edge instabilities known as edge localized modes (ELMs) [10, 11]"

However, we would like to argue our original point that canceling the $n=1$ field may not be favorable for ELM suppression using $n=3$ or $n=4$ RMPs. This perspective is supported by previous ELM suppression experiments using mixed toroidal harmonic RMP in DIII-D, which demonstrated an approximately 35% reduction in the $n=3$ RMP threshold with the application of $n=2$ RMPs [S. Gu et al., NF 2019], as depicted in Fig. B. This suggests that the presence of the intrinsic $n=1$ field could potentially reduce the required $n=3$ or $n=4$ RMP coil currents for ELM suppression, thus, we used the term "contrary" for this purpose in the previous version of the manuscript.

Figure B. ELM suppression threshold measured by ramping up single harmonic and mixed toroidal harmonic RMP current. Temporal evolution of (a) amplitude of the single harmonic $n=3$ in pulse 170077 (blue dashed line) and the mixed toroidal $n=2$ (red solid line) and $n=3$ (red dash-dot line) harmonic in pulse 170079, (b) D_α emission (blue solid line) in the single harmonic pulse 170077, (c) D_α emission (red solid line) in the mixed toroidal harmonic pulse 170079, and (d) density on pedestal in pulse 170077 (blue dashed line) and 170079 (red solid line). Reproduced/reprinted with permission from [S. Gu et al. Edge localized mode suppression and plasma response using mixed toroidal harmonic resonant magnetic perturbations in DIII-D. Nucl. Fusion 59 026012 (2019)]

Comment 2) *Figure 1a. What is the scale in the colour scheme for the perturbation? Does 1a really show the displacement of the flux surfaces (how I would interpret the term “perturbed flux surface”) or the perturbation of the magnetic field as in 1b?*

Response to comment 2:

Yes, it does depict displacement on the flux surface. However, as pointed out by the reviewer, the distortion is not big enough to be visible in Fig. 1a, primarily due to the weak plasma response at low beta. Please refer to the modified description in Fig. 1a described below.

Page 3: “The actual geometry of 3D coils and perturbed flux surface due to the plasma response (perturbation is not big enough to see the distortion of flux surface in this figure)”

Comment 3) *Figure 1b: What is the definition of δB_n^x ?*

Response to comment 3:

Thank you for pointing this out. It is an externally applied normal magnetic field at the plasma boundary. Please refer to the modified description in Fig. 1b.

Page 3: “The externally applied normal magnetic field at the plasma boundary (δB_n^x)”

Comment 4) *Line 49. Instead of nuclear contamination you must be thinking of nuclear degradation of the coils.*

Response to comment 4:

Thank you for picking this up. We corrected this in line 48.

Lines 47-48: “However, future tokamak reactors such as DEMO will need ex-vessel 3D coils to avoid nuclear degradation of the coils.”

Comment 5) Line 80. *The term resonant should be defined. It is used a lot in the rest of the paper. What is resonating with what?*

Response to comment 6:

Thank you again for pointing this out. In the modified manuscript, we have explained the term "resonant." Please refer to lines 83 to 86.

Lines 83-86: "The term "resonant" in this work represents a 3D field that resonates with an equilibrium magnetic field. More precisely, the resonant components are defined as the $m = nq$ harmonics at rational surfaces, where magnetic field lines close in on themselves after m toroidal transits and n poloidal transits. Here, q is the so-called safety factor."

Comment 7) Line 90. *D_α emission does not indicate the hard sputtering of deuterium ions, but is due to ionization of neutral gas in the divertor region.*

Response to comment 7:

Thank you for this correction. We corrected this in the modified manuscript (please refer to line 96).

Lines 96-97: "Here, the repetitive spikes in D_α emission indicate the ionization of neutral gas in the divertor region due to type-I ELM crashes,"

Comment 8) 107. *This relates to my general comment. The threshold is mentioned here the first time. It should be defined and in particular it should be explained how the threshold is used in the following modelling.*

Response to comment 8:

Thank you once again. As addressed in the 'Response to general comment 2,' we have defined and provided an additional explanation of the threshold in the modified manuscript. Please also see the sentences from lines 372 to 388 in the modified manuscript.

Comment 9) Figure 4. *The letters and the caption descriptions do not match. The caption description does not even say anything about f and g plots. Also d is obviously not "NTV torque profile" but most likely density profile.*

Response to comment 9:

We appreciate the reviewer for picking up this mistake. We corrected the caption description of Figure 4 in the modified manuscript as shown below.

Figure 4: Comparison of transport with different 3D field spectra. **a**, Time traces of plasma current I_p , root sum squared coil current amplitude I_{3D} , and D_α emission with different 3D field spectra. **b**, edge normalized resonant field profiles ($\delta B_{\text{res}}/\delta B_{\text{res}}(\psi_N \sim 0.95)$) with a different 3D field spectrum. **c,d**, Toroidal rotation frequency and density profiles before 3D fields at $t = 5.00\text{s} \pm 0.02\text{s}$. **e**, Calculated Neoclassical Toroidal Viscosity (NTV) torque profiles. **f,g**, Toroidal rotation frequency and density profiles after ELM suppression at $t = 9.06\text{s} \pm 0.02\text{s}$, $t = 8.13 \pm 0.02\text{s}$, and $t = 7.47 \pm 0.02\text{s}$ of discharge, #26016 (green), #26014 (blue), #26015 (red) respectively. The selected time points are indicated with a vertical dashed line in **a**.

Comment 10) Last paragraph of page 8: Is there an explanation why the plasma response increases the Chirikov parameter in one case and decreases it in the other?

Response to comment 10:

This is because the two applied 3D fields have different poloidal spectra, and the plasma responds differently to these spectra. The plasma exhibits a higher sensitivity to specific components or distributions of the applied 3D field at the plasma boundary. Due to the plasma response, including poloidal coupling and amplification resulting from stable kink modes, the results significantly deviate from the predictions made under the vacuum approximation. Please refer to lines 177-179 in the modified manuscript.

Lines 177-179: "As two EFCs have different poloidal spectra, the experiment can investigate the reliability of the predicted plasma response model, which exhibits a higher sensitivity to specific components or distributions of the applied 3D field."

Comment 11) Line 189-190: What sustains higher ion temperature gradient in the ERMP case?

Response to comment 11:

We believe a high fraction of fast ions plays a crucial role in establishing this higher ion temperature gradient (lines 201-202 in the modified manuscript). A more detailed analysis of this higher ion temperature gradient scenario in KSTAR is reported in [Han et al., Nature 2022].

Lines 199-202: “These results show that H-mode avoidance using an $n = 1$ ERMP not only eliminates the uncertainties in ELM destabilization but also helps long and robust sustainment of improved core confinement with $T_i \approx 100$ million kelvin **by maintaining high fast particle fraction [38]**”

REVIEWERS' COMMENTS

Reviewer #1 (Remarks to the Author):

The author has addressed my remarks satisfactorily. thank you.

Reviewer #2 (Remarks to the Author):

The authors answered properly all of my comments. The optimization scheme might be practically useful for fusion application, although it is based on known linear plasma response physics. I agree the publication of this paper in present form.

Reviewer #3 (Remarks to the Author):

The authors have adequately responded to all my comments and made changes to the manuscript so that I have no further comments on it. I recommend publishing it in Nature Communications.

RESPONSES TO REFEREE #1's COMMENTS:

The author has addressed my remarks satisfactorily. thank you.

We sincerely thank the referee's previous support and helpful comments, which have been very helpful for us in clarifying the issues and improving the presentations of our work.

RESPONSES TO REFEREE #2's COMMENTS:

The authors answered properly all of my comments. The optimization scheme might be practically useful for fusion application, although it is based on known linear plasma response physics. I agree the publication of this paper in present form.

We deeply appreciate the referee's previous comments, which led to a considerable improvement in the paper. Thanks for the referee's agreement for publication in Nature Communications.

RESPONSES TO REFEREE #3's COMMENTS:

The authors have adequately responded to all my comments and made changes to the manuscript so that I have no further comments on it. I recommend publishing it in Nature Communications.

We are sincerely grateful for the referee's previous fruitful comments that led to a significant improvement of the manuscript. Thanks for the referee's warm recommendation for publication in Nature Communications.